# The Changes in Bell Pepper Flesh as a Result of Lacto-Fermentation Evaluated Using Image Features and Machine Learning

**DOI:** 10.3390/foods11192956

**Published:** 2022-09-21

**Authors:** Ewa Ropelewska, Kadir Sabanci, Muhammet Fatih Aslan

**Affiliations:** 1Fruit and Vegetable Storage and Processing Department, The National Institute of Horticultural Research, Konstytucji 3 Maja 1/3, 96-100 Skierniewice, Poland; 2Department of Electrical and Electronics Engineering, Karamanoglu Mehmetbey University, Karaman 70100, Turkey

**Keywords:** pepper preservation, spontaneous lacto-fermentation, image processing, texture parameters, discrimination, machine learning algorithms

## Abstract

Food processing allows for maintaining the quality of perishable products and extending their shelf life. Nondestructive procedures combining image analysis and machine learning can be used to control the quality of processed foods. This study was aimed at developing an innovative approach to distinguishing fresh and lacto-fermented red bell pepper samples involving selected image textures and machine learning algorithms. Before processing, the pieces of fresh pepper and samples subjected to spontaneous lacto-fermentation were imaged using a digital camera. The texture parameters were extracted from images converted to different color channels *L*, *a*, *b*, *R*, *G*, *B*, *X*, *Y*, and *Z*. The textures after selection were used to build models for the classification of fresh and lacto-fermented samples using algorithms from the groups of Lazy, Functions, Trees, Bayes, Meta, and Rules. The highest average accuracy of classification reached 99% for the models developed based on sets of selected textures for color space Lab using the IBk (instance-based K-nearest learner) algorithm from the group of Lazy, color space RGB using SMO (sequential minimal optimization) from Functions, and color space XYZ and color channel *X* using IBk (Lazy) and SMO (Functions). The results confirmed the differences in image features of fresh and lacto-fermented red bell pepper and revealed the effectiveness of models built based on textures using machine learning algorithms for the evaluation of the changes in the pepper flesh structure caused by processing.

## 1. Introduction

Bell pepper (*Capsicum annum* L.), belonging to Solanaceae, is a widely cultivated fruit, used as a vegetable, spice, or condiment. Consuming pepper can provide health benefits due to the presence of phytochemicals, including phenolic compounds, capsaicinoids, vitamins C and E, and carotenoids [1]. Processing of pepper fruit can prolong the storage time and provide value-added food products [1]. Strong antioxidant capacity, high content of bioactive compounds, distinctive colors, flavor, and nutritional value result in the popularity of consuming bell pepper. Bell pepper color may be related to significant differences in taste, antioxidant capacity, bioactive compounds, nutrient content, antioxidant capacity, and cost. Green bell pepper contains chlorophylls and distinctive carotenoids (lutein, neoxanthin, and violaxanthin) [2]. Red pepper is characterized by the presence of capsorubin and capsanthin [2]. In the case of yellow pepper, violaxanthin, β-carotene, lutein, zeaxanthin, and antheraxanthin are the most common. The higher concentrations of polyphenols, among others, flavonoid and quercetin, are determined in red and yellow peppers compared to green fruit [2]. Due to the presence mainly of carotenoids, flavonoids, and vitamins, bell pepper is an important ingredient against aging and preventing chronic diseases and can be used as a food and medicinally [3,4]. Bell pepper is characterized by short storage and shelf-life, and it is a non-climacteric fruit insensitive to ethylene but sensitive to cold temperature. Pepper due to peel without stomata is a special fruit. Water loss through the cuticle is the only source of shriveling, as well as weight and water loss [5]. The deterioration of products can appear at any stage of the supply chain. Therefore, developing approaches to the preservation of food products, allowing for their longer use, including packing and processing, is important [6].

People are interested in consuming the fresh-cut bell pepper without loss of nutritional value. The transportation of fresh-cut pepper products can be more voluminous than that of whole bell pepper in terms of kilograms, using the same container. Cutting a pepper can reduce the transportation charges and increase the economic value. However, cutting bell pepper may cause a risk of pathogen contamination, tissue damage, loss of flavor and texture, browning, and the release of nutrients. To inhibit microbial contamination, processing the fresh-cut pepper can be applied [7]. In addition to fresh, pepper can be eaten in various processed forms, e.g., fermented, dried, cooked, or as extracts [8]. Lacto-fermentation is one of the traditional ways of fruit preservation and processing. Fermentation represents cultural heritage and provides regional products. Fermented pepper, among others, can prevent fat accumulation and reduce lipid levels [9]. Additionally, lacto-fermentation can enhance antimicrobial and antioxidant activity. Fermentation involving selected strains of lactic acid bacteria (LAB) can release bioactive compounds from cells and enhance biological activity [10]. Fermentation using probiotic bacteria can also enhance the shelf life, flavor, sensory and nutritional attributes, bioavailability, and functional quality of products [11,12].

The effect caused by lacto-fermentation on red bell peppers can be observed by an expert via conventional measurement and observation. However, with Agriculture 4.0, manual and cumbersome agricultural solutions are now being replaced by computer-based automatic systems. Currently, the application of noninvasive measuring techniques to evaluate external and internal fruit quality, e.g., involving image processing and artificial intelligence, is desired in the food industry [13]. Machine learning includes algorithms and tools needed to understand patterns within the data, extract important information from the data, and make decisions on a specific task by machines [14]. Thanks to computer vision systems, many precision agriculture applications such as monitoring of plant diseases, detection of pests, detection of fruit/vegetables, spraying, and mapping are carried out automatically and quickly [15]. In order for these applications to be carried out automatically, the data obtained from different sensors is analyzed by a computer instead of an expert. However, most of the time, these data are not used directly, and preprocessing is performed on the raw data. Image data are mostly obtained in agricultural applications, and, in this sense, image processing techniques are of great importance. Preprocessing and feature extraction algorithms in image processing have been used in many agricultural applications [16]. With image preprocessing, the raw image is transformed into a suitable one for the desired task, and different steps such as thresholding, morphological operations, segmentation, etc. are applied to the raw image [17]. Then, different features related to texture, shape, and color are extracted from the processed image with feature extraction steps [18]. These features are fed into learning algorithms to automatically provide the desired discrimination. In this context, artificial intelligence-based methods such as machine learning and deep learning are used to automatically distinguish different species on the basis of these images or these features [19]. In agricultural discrimination applications, texture features are mostly used because of their strong discrimination ability [20]. MaZda, an open source C++ library, is a highly preferred software package as it allows both 2D and 3D image texture analysis and image preprocessing [21]. For this reason, it has been used in numerous agricultural applications. The change caused by lacto-fermentation can also be observed by analyzing textural features through MaZda. In addition, it is very important and necessary to automatically determine the effect of lacto-fermentation, as it is often applied for different fruits and vegetables. The image features and machine learning algorithms were successfully used in previous studies for evaluation of the changes, e.g., in carrot [22], cucumber [23], and beetroot [24,25], caused by lacto-fermentation.

This study offers an innovative and comprehensive approach to assessing the quality of bell pepper fruit. In this context, the supplied red bell pepper sample pieces are imaged before and after lacto-fermentation. Each of these images is converted to *L*, *a*, *b*, *R*, *G*, *B*, *X*, *Y*, and *Z* color channels. Using the brightness threshold, each sample is segmented from the background, and regions of interest (ROI) are generated. Then, various texture features are extracted from ROI images with different color channels. Texture features extracted from different color channels show the effect of different textural information on the discrimination process. Feature selection algorithms are used for a large number of texture features obtained at this stage. Finally, these selected features are analyzed using various machine learning algorithms, resulting in highly accurate discrimination of fresh and lacto-fermented red bell pepper samples. The results show that the proposed method is capable of distinguishing with an accuracy of up to 99%.

Despite the published results, mainly for other species of fruit and vegetables, there are no literature data on the use of texture parameters extracted from various color channels of images of cut red bell pepper to build classification models for monitoring the effect of lacto-fermentation on changes in the structure of pepper flesh. Therefore, the objective of this study was to propose an innovative approach to distinguishing fresh and lacto-fermented red bell pepper samples involving selected texture parameters of images and various machine learning algorithms. The application of textures from different color channels *L*, *a*, *b*, *R*, *G*, *B*, *X*, *Y*, and *Z* and algorithms belonging to different groups to evaluate the changes in red bell pepper flesh occurring as a result of spontaneous lacto-fermentation is a great novelty of this study. The innovative nature of the present study involves the acquisition of data on more than 1600 texture parameters of bell pepper flesh in the fresh and lacto-fermented forms. Various methods of selecting textures with the highest discriminant power, such as genetic search and best first with the correlation-based feature selection (CFS), as well as the ranker in conjunction with OneR attribute evaluator, were used to choose the most effective one. A wide range of algorithms from the groups of Lazy, Functions, Trees, Bayes, Meta, and Rules were tested, which was not found in the literature data for the classification of fresh and processed bell pepper. The use of various algorithms in the discrimination process shows the validity, applicability, and robustness of the proposed method rather than making a comparison between machine learning algorithms.

This paper is organized as follows: after the introduction, Section 2 deals with the acquisition, imaging, processing, and statistical analysis of bell pepper samples. Section 3 includes the discrimination performances obtained as a result of evaluating the texture features extracted from different color channels with different machine learning algorithms. Lastly, Section 4 evaluates the proposed work and provides suggestions for future work.

## 2. Materials and Methods

### 2.1. Materials

The red bell pepper samples (Figure 1) were bought at a supermarket. A total of 20 mature undamaged fruits were selected for this study. Fruits were washed and cleaned. Then, five pieces with the dimensions of 1 cm × 1 cm were cut from each pepper fruit using a sharp stainless-steel knife. The extracted fresh pepper pieces were subjected to imaging. The same pieces were also intended for spontaneous lacto-fermentation and then imaging as lacto-fermented forms.

The spontaneous lacto-fermentation using garlic, horseradish, dill, and potable water with table salt at the final concentration of 3% sodium chloride in brine was applied. The previously prepared red bell pepper pieces were put into glass jars with other ingredients. Samples were stored for 3 days at a temperature of about 20 °C and then for 6 months at about 10–12 °C. After storage, lacto-fermented pepper samples were rinsed under potable water and dried with a paper towel. The samples prepared in this way were subjected to imaging using a digital camera.

### 2.2. Image Acquisition and Processing

The same imaging procedure was applied for the pieces of fresh pepper before processing and lacto-fermented pepper. The samples were imaged on the inside of the flesh using a digital camera placed on a tripod in a box with black internal walls. As a light source, LED (light-emitting diode) illumination with stable parameters was used. Color calibration of the digital camera was carried out. The acquired images *contained* red bell pepper pieces on a black background. There were 20 pepper pieces in one image. In total, images of 100 pieces of fresh pepper and 100 pieces of lacto-fermented pepper were obtained. The image processing using the MaZda application (Łódź University of Technology, Institute of Electronics, Łódź, Poland) [21,26,27] allowed computing texture parameters from different color channels *L*, *a*, *b*, *R*, *G*, *B*, *X*, *Y*, and *Z*. Firstly, the fresh and lacto-fermented bell pepper images were converted to color channels. In the case of lacto-fermented samples, changes in the structure of the flesh compared to the fresh samples are visible (Figure 2).

The regions of interest (ROIs) were overlaid. An ROI was considered as a set of pixels separated from the background. The segmentation of the image into lighter pepper pieces and the black background was performed using the brightness threshold which was determined manually. Each ROI was one whole piece of pepper. Thus, 200 ROIs including 100 ROIs of fresh samples and 100 ROIs of lacto-fermented samples were determined. For each ROI, 1629 image textures were computed including 181 textures for each color channel. The texture parameters were computed on the basis of the co-occurrence matrix (132 textures including 11 features for four directions and three between-pixel distances), run-length matrix (20 textures including five textures for four directions), Haar wavelet transform (10 textures), histogram (nine textures), autoregressive model (five textures), ad gradient map (five textures). Among the color channels of images selected for the texture extraction, color channels *R* (red), *G* (green), and *B* (blue) belonged to the RGB color space, channels *L* (lightness from black to white), *a* (green for negative and red for positive values), and *b* (blue for negative and yellow for positive values) belonged to the Lab color space, and channels *X* (a component of color information), *Y* (lightness), and *Z* (a component of color information) belonged to the XYZ color space [28]. The computed image textures were considered as a function of the spatial variation of the pixel brightness intensity and provided information about the structure of the samples. Thus, the quantitative analysis of textures provided important insights into sample quality. The texture parameters after selection were used to build the discriminative models for distinguishing fresh and lacto-fermented red bell pepper samples. A flowchart including steps of the applied procedure is presented in Figure 3.

### 2.3. Statistical Analysis

Statistical analysis involved the development of innovative models for distinguishing fresh and lacto-fermented red bell pepper samples on the basis of image features. The discriminant analysis was performed using WEKA software (Machine Learning Group, University of Waikato) [29,30,31] and included several steps. In the first step, the texture selection was performed. This procedure was applied for sets of textures computed for color spaces Lab, RGB, and XYZ including combined textures for three channels in the case of each color space, and for individual color channels *L*, *a*, *b*, *R*, *G*, *B*, *X*, *Y*, and *Z*. In the case of each color space, the most satisfactory results for one channel were chosen to be presented in this paper. Among the search methods, genetic search and best first with the correlation-based feature selection (CFS), and the ranker in conjunction with OneR attribute evaluator were applied. Various algorithms from different groups were tested such as Lazy (LWL—locally weighted learning, KStar, IBk—instance-based K-nearest learner), Functions (logistic, LDA—linear discriminant analysis, FLDA—Fisher linear discriminant analysis, QDA—quadratic discriminant analysis, SMO—sequential minimal optimization), Trees (LMT—logistic model tree, random forest, J48), Bayes (Bayes net, naïve Bayes), Meta (multi class classifier, filtered classifier, random committee, logit boost), and Rules (JRip—Java repeated incremental pruning, PART). A test mode of 10-fold cross-validation was used to perform the analysis. The dataset including a total of 200 cases was randomly divided into 10 parts. Each of the ten parts was considered in turn and treated as the test set, whereas the remaining nine parts were regarded as the training sets. The learning was performed 10 times using different training sets. The result was determined as the average of 10 estimates. Ten folds are generally sufficient to obtain the best estimate [31]. The criterion for the selection of machine learning algorithms and evaluation of analysis was the highest average accuracy of classification. In addition to average accuracy and confusion matrices including accuracies for fresh and lacto-fermented samples, other performance metrics, such as precision, F-measure, and MCC (Matthews correlation coefficient), were also determined [32,33].

## 3. Results

The innovative models based on selected image textures for distinguishing fresh and lacto-fermented red bell pepper samples were developed using machine learning algorithms. The models were built separately for color spaces and color channels. From the tested search methods, best first with CFS proved to be the most satisfactory in terms of sets of selected textures providing the highest accuracies. The textures selected using the best first with CFS are presented in Table 1. It was observed that the highest results were ensured by models developed using the following machine learning algorithms: IBk (Lazy), SMO (Functions), random forest (Trees), naïive Bayes (Bayes), filtered classifier (Meta), and JRip (Rules).

In the case of the models built on the basis of selected textures from the color space Lab (Table 2), an average accuracy of discrimination of fresh and lacto-fermented pepper pieces reached 99% for the Ibk algorithm from the group of Lazy. Only this algorithm provided 100% accuracy for one of the classes (lacto-fermented). The fresh pepper samples were discriminated with an accuracy of 98%, and the remaining 2% were incorrectly included in the class ‘lacto-fermented’. Other discrimination performance metrics were also the most satisfactory for the model built using IBk. The values of precision reached 1.000 for fresh pepper samples and 0.980 for lacto-fermented samples, whereas both classes were characterized by an F-measure parameter equal to 0.990 and MCC equal to 0.980. For the other algorithms, high values of metrics were also obtained. The models built using the SMO algorithm from the group of Functions, random forest from Trees, and naïve Bayes from Bayes correctly discriminated fresh and lacto-fermented pepper samples in 98% of cases, while other metrics were greater than or equal to 0.960, reaching 0.990 for precision for fresh samples and random forest for lacto-fermented samples, as well as naïve Bayes. Slightly lower average accuracies were observed for filtered classifier from the group of Meta (97%) and Jrip from the group of Rules (96.5%).

In the case of Lab color space, among the individual color channels, models including textures selected from color channel *L* (Table 3) provided the most satisfactory results. The fresh and lacto-fermented pepper samples were correctly distinguished from each other with an average accuracy of up to 98.5% for the random forest. The fresh pepper samples were classified with an accuracy of 98%, whereas lacto-fermented pepper samples were classified with an accuracy of 99%. For models developed using other machine learning algorithms, very high average accuracies were also found: 98% for IBk and SMO, 97% for naïve Bayes and filtered classifier, and 96.5% for JRip. The accuracy of 100% was not observed for any of the classes. The highest value of precision of 0.990 was determined for the fresh pepper and random forest. The F-measure of 0.985 and MCC of 0.970 for both classes were also the highest for the random forest algorithm.

Furthermore, high average accuracies reaching 99% (SMO algorithm) were determined for models including image textures selected for RGB color space (Table 4). For SMO, all cases belonging to fresh pepper were correctly classified as fresh pepper (100% accuracy), and 98% of cases from the actual class ‘lacto-fermented’ were correctly included in the predicted class ‘lacto-fermented’. The values of precision (0.980 for fresh pepper samples and 1.000 for lacto-fermented pepper samples), F-measure (0.990 for both classes), and MCC (0.980 for both classes) were also very satisfactory. Moreover, a high average accuracy of 98.5% was observed for the IBk and naïve Bayes machine learning algorithms. However, in the case of some models, the average accuracies were lower, such as 93% for the model built using filtered classifier and 93.5% for the model developed using JRip.

From color channels belonging to RGB color space, the highest results of discrimination of fresh and lacto-fermented red bell pepper samples were obtained for models built on the basis of selected image textures from channel *R* (Table 5). An average accuracy of up to 98.5% (SMO) was determined. In the case of the SMO algorithm which provided the most satisfactory results, the discrimination accuracy for fresh pepper was equal to 99%, and 1% of samples were incorrectly classified as lacto-fermented pepper, whereas lacto-fermented samples were correctly discriminated in 98% of cases, and the remaining 2% were incorrectly included in the predicted class ‘fresh pepper’. The values of precision reached 0.980 and 0.990, for fresh and lacto-fermented samples, respectively. The highest F-measure equal to 0.985 and MCC equal to 0.970 were found for both classes. Among the other algorithms, Ibk (97.5%), random forest (97%), and naïve Bayes (96%) also allowed for building effective models. In the case of filtered classifier and JRip, lower average accuracies (91.5% and 90.5%, respectively) were determined.

Very high results were also produced by the models built on the basis of selected textures from the XYZ color space (Table 6). The average accuracy reached 99% in the case of the models built using the IBk and SMO machine learning algorithms. A slightly lower average accuracy of 98% was obtained for random forest and naïve Bayes. Furthermore, for filtered classifier and JRip, average accuracies were satisfactory (97% and 96.5%, respectively). In the case of individual classes, 100% accuracy was observed for fresh pepper for the SMO algorithm, whereas lacto-fermented cases were correctly distinguished from fresh samples in 98% of cases. The highest results of precision, F-measure, and MCC were determined for the models built using the IBk and SMO algorithms. In the case of IBk, precision reached 0.990 for both classes, while, for SMO, the value of precision was equal to 1.000 for the lacto-fermented pepper and 0.980 for the fresh pepper. Both for IBk and SMO, F-measure reaching 0.990 and an MCC of 0.980 for both classes were obtained.

For models developed using selected textures from images converted to the color channel *X* (Table 7), the same results as for color space XYZ (Table 6) were observed for the IBk and SMO algorithms. The average accuracy also reached 99% (IBk, SMO), and an accuracy of 100% was only found for fresh pepper (SMO). In the case of color channel *X* (Table 7), for other algorithms, fresh lacto-fermented pepper samples were correctly distinguished from each other in 98.5% of cases for naïve Bayes, 97% of cases for random forest, 95.5% of cases for filtered classifier, and 93.5% of cases for JRip.

When Table 2, Table 3, Table 4, Table 5, Table 6 and Table 7, which contain the results of this study, are examined, the findings regarding the experimental results can be expressed as follows: with image analysis, artificial intelligence, and computer vision algorithms, fresh and lacto-fermented red bell pepper samples were distinguished in an objective, nondestructive, and inexpensive way. In this way, the quality of the crop was determined automatically, quickly, and without bias [34]. Furthermore, the texture features of the different color channels used (*L*, *a*, *b*, *R*, *G*, *B*, *X*, *Y*, and *Z*) strongly represented the changes in lacto-fermented red bell pepper samples. It was confirmed that these texture features can also be strongly distinguished by various machine learning algorithms from different groups (Lazy, Functions, Trees, Bayes, Meta, and Rules). According to the experimental results, the most erroneous discrimination was 90.5% with the Jrip algorithm, while the most successful discrimination was 99% with the IBk algorithm. Therefore, these results showed that the proposed method is highly preferable over conventional methods. Moreover, the results determined for red bell pepper in the present study confirmed the previous literature data on the effectiveness of models built based on image parameters using machine learning algorithms for the evaluation of the quality of fermented food products such as cucumber, carrot, or beetroot [22,23,24,25]. The undertaken research extends the scope of application of image analysis and artificial intelligence to the evaluation of fruit and vegetables by assessing the quality of fermented products.

Image processing and traditional machine learning and deep learning are also widely used for other fruit and vegetable research and agriculture activities. Machine learning techniques and algorithms can be used in the pre-harvesting, harvesting, and post-harvesting stages [34]. This state-of-art technology can help to solve problems in agriculture and help farmers to reduce losses. In the pre-harvesting stage, machine learning can be applied, for example, to evaluate the quality and germination of the seeds, sort seeds, detect disease and weeds, capture the parameters of soil, pruning, fertilizer application, irrigation, and determining genetic and environmental conditions. In the harvesting stage, the activities involving machine learning may be related to crop size, quality, skin color, taste, maturity stage, firmness, market window, object detection, and classification [35]. On the other hand, post-harvesting applying machine learning may concern factors affecting the shelf-life, e.g., temperature, gases used in containers, humidity, usage of chemicals, handling processes to quality retain, and grading [35]. In the case of pepper, machine learning was used for nondestructive sorting based on odor parameters [36]. Models developed using a deep convolutional neural network (DCNN) were applied by Subeesh et al. [14] for weed detection in the polyhouse cultivation of bell peppers. Mohi-Alden et al. [37] used machine vision intelligent modeling for in-line sorting of bell pepper. Red and yellow sweet peppers were classified into immature and mature classes on the basis of color and morphological features using machine learning algorithms [38]. Additionally, the ripeness level was estimated on the basis of color image parameters using machine learning models in the case of grapes [39]. Image analysis, spectroscopy, and electronic nose combined with different statistical models were applied to discriminate the ripening stage of strawberries [40]. Models based on spectral reflectance data built using machine learning algorithms were used to distinguish the *Fusarium*-infected and healthy pepper samples (leaves) [41]. In view of the promising application of objective and nondestructive procedures involving artificial intelligence and image processing to evaluate fruit and vegetables, further research may include other species and cultivars, as well as novel directions of experiments.

## 4. Conclusions

Preservation and processing of bell peppers by lacto-fermentation is a traditional method to extend their shelf life. In this context, the main justification for this study was to evaluate the effect of the lacto-fermentation method on red bell peppers. The approach involving image analysis and machine learning enabled distinguishing fresh and lacto-fermented red bell pepper samples (flesh pieces) on the basis of models including selected textures built using various algorithms. Models developed for sets of textures selected separately for color spaces and color channels provided satisfactory results. The highest results, including the average discrimination accuracy reaching 99%, were obtained for IBk (Lazy), SMO (Functions), random forest (Trees), naïve Bayes (Bayes), filtered classifier (Meta), and JRip (Rules) machine learning algorithms. The high discrimination accuracies, as well as the values of other metrics, such as precision, F-measure, and MCC (Matthews correlation coefficient), revealed differentiation of fresh and lacto-fermented bell pepper samples. Therefore, the changes in the bell pepper flesh structure caused by spontaneous lacto-fermentation were confirmed. Due to the evaluation of the quality of lacto-fermented bell pepper using textures from different color channels *L*, *a*, *b*, *R*, *G*, *B*, *X*, *Y*, and *Z* and various machine learning algorithms belonging to different groups, this study is characterized by novelty. The obtained results are very promising. Therefore, further research on the evaluation of the effect of lacto-fermentation on the flesh of other cultivars of pepper and other species of fruit or vegetables may be performed.

The texture features extracted from the different color channels strongly indicated the effect of lacto fermentation. However, a single texture feature may not be sufficient to distinguish lacto-fermented red bell pepper samples. The features extracted in machine learning algorithms determine the performance of the system. The features should, therefore, strongly represent the distinction between species. Therefore, in future studies, more than one texture feature will be fused, and the effect of lacto-fermentation will be revealed more strongly. In addition, the new trend for agricultural discrimination tasks is to develop deep learning-based methods. Accordingly, more powerful features are automatically extracted instead of manual feature extraction. Therefore, deep learning-based techniques can provide easier and more accurate discrimination. However, to see the superior performance of deep learning clearly, more samples are required than machine learning. Therefore, in our next studies, the effect of lacto-fermentation will be analyzed with deep learning-based CNN models that enable the extraction of high-level features. For this, datasets containing more samples will be prepared. Lastly, the proposed study is only for bell pepper and, therefore, does not produce good results in differentiating different vegetables or fruits as a result of lacto fermentation. In this sense, it is planned to create a large dataset containing different fruits and vegetables in future studies.

## Figures and Tables

**Figure 1 foods-11-02956-f001:**
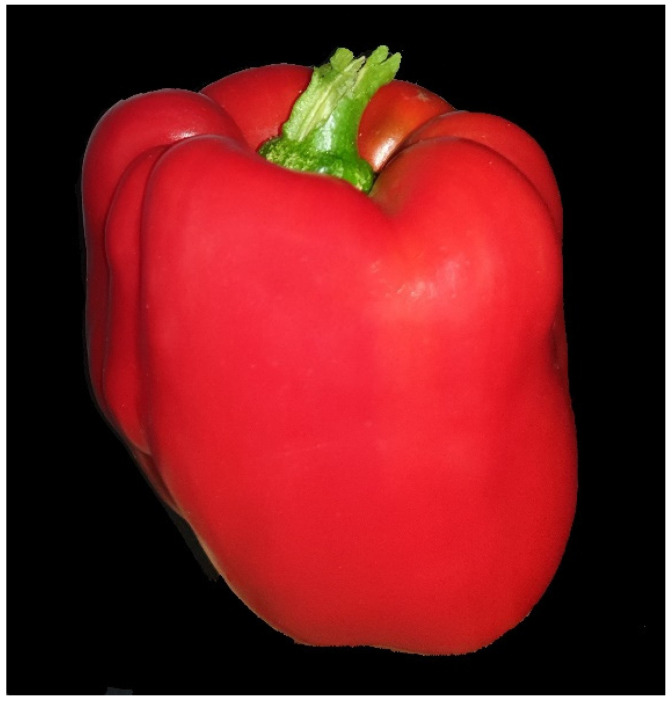
The sample of red bell pepper fruit used in the experiments.

**Figure 2 foods-11-02956-f002:**
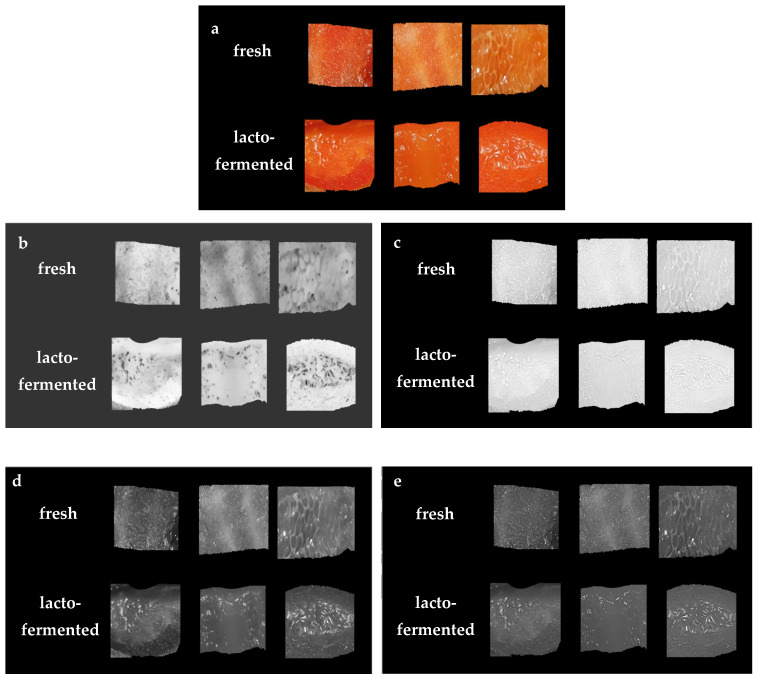
The original digital color images of samples of fresh and lacto-fermented bell pepper (**a**) and images converted to selected color channels: *a* (**b**); *R* (**c**); *G* (**d**); *X* (**e**).

**Figure 3 foods-11-02956-f003:**
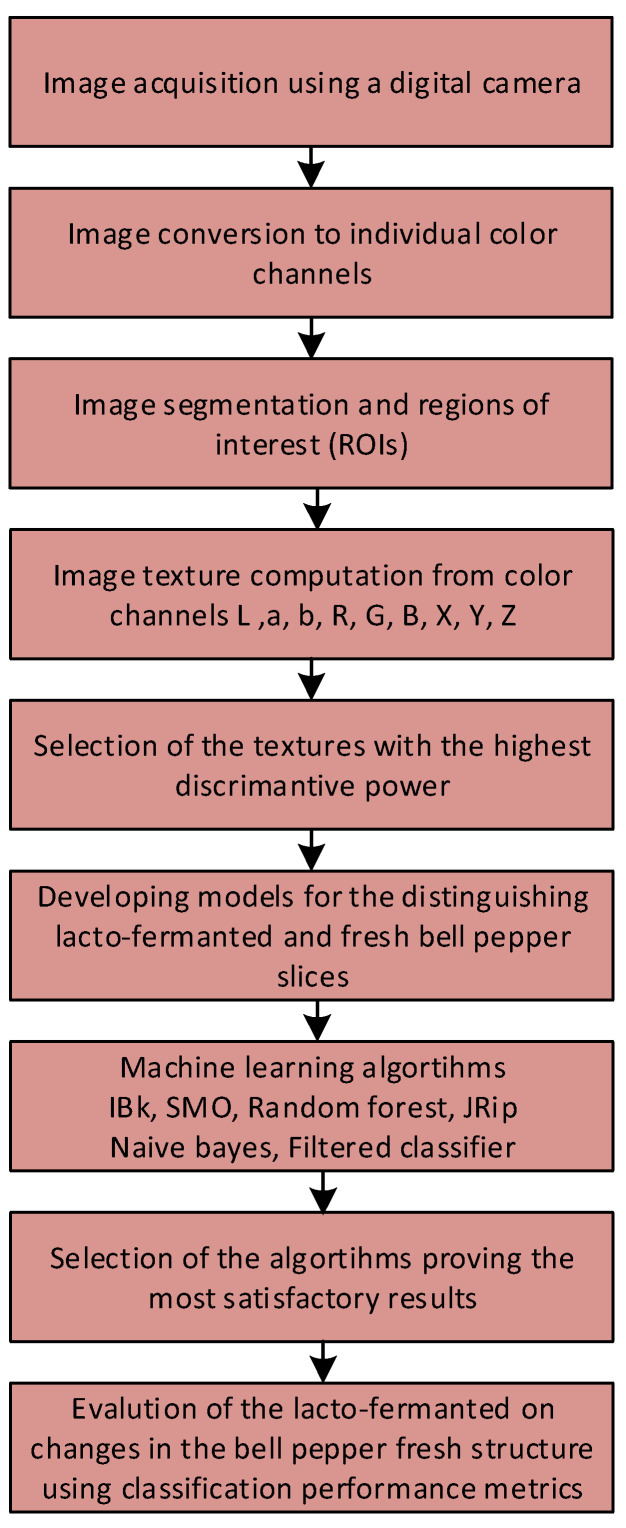
A flowchart presenting steps of distinguishing fresh and lacto-fermented red bell pepper samples using image textures and machine learning algorithms.

**Table 1 foods-11-02956-t001:** The selected texture parameters used to develop the models to distinguish fresh and lacto-fermented red bell pepper samples.

Color Space Lab	Color Channel *L*	Color Space RGB	Color Channel *R*	Color Space XYZ	Color Channel *X*
LHMean LHPerc01 LHPerc10 LHMaxm10 LS5SH1SumEntrp LS5SH3Correlat LS5SV5AngScMom LS4RZGLevNonU aS5SZ1Correlat aS4RNRLNonUni bHDomn01 bS5SN1AngScMom bS4RHShrtREmp bS4RNLngREmph	LHMean LHPerc01 LHPerc10 LHPerc99 LHMaxm10 LS5SH1SumEntrp LS5SH3Correlat LS5SV5AngScMom LS4RZGLevNonU LS4RNRLNonUni	RHPerc10 RHMaxm01 RHMaxm10 RSGVariance RS5SH1Correlat RS5SH3Correlat RS5SH3SumVarnc RS5SH5Correlat RS5SV5Correlat RS4RZGLevNonU RATeta1 GHKurtosis GHPerc01 GHPerc50 GHMaxm10 GS5SH1SumEntrp GS5SH3SumVarnc GS5SN5AngScMom GS4RNRLNonUni BHDomn10	RHPerc10 RHMaxm01 RHMaxm10 RS5SH1Correlat RS5SZ1DifVarnc RS5SH3Correlat RS5SN3AngScMom RS5SH5Correlat RS5SV5Correlat RS4RHGLevNonU RATeta1	XHPerc10 XHMaxm01 XHMaxm10 XS5SH1Correlat XS5SV1SumAverg XS5SH3SumEntrp XS5SZ3SumEntrp XS5SH5Correlat XS4RVGLevNonU XASigma YHDomn01 YHMaxm10 YS5SZ1SumEntrp YS4RNRLNonUni ZSGNonZeros	XHPerc10 XHMaxm01 XHMaxm10 XS5SH1Correlat XS5SV1SumAverg XS5SH3SumEntrp XS5SZ3SumEntrp XS5SH5Correlat XS4RVGLevNonU XS4RNGLevNonU XASigma

**Table 2 foods-11-02956-t002:** The performance metrics of discrimination of fresh and lacto-fermented red bell pepper samples using selected image textures from color space Lab.

Algorithm (Group)	Predicted Class (%)		Actual Class	Average Accuracy (%)	Precision	F-Measure	MCC
	Fresh	Lacto-Fermented					
IBk (Lazy)	98	2	Fresh	99	1.000	0.990	0.980
	0	100	Lacto-fermented		0.980	0.990	0.980
SMO (Functions)	98	2	Fresh	98	0.980	0.980	0.960
	2	98	Lacto-fermented		0.980	0.980	0.960
Random forest (Trees)	97	3	Fresh	98	0.990	0.980	0.960
	1	99	Lacto-fermented		0.971	0.980	0.960
Naïve Bayes (Bayes)	99	1	Fresh	98	0.971	0.980	0.960
	3	97	Lacto-fermented		0.990	0.980	0.960
Filtered classifier (Meta)	97	3	Fresh	97	0.970	0.970	0.940
	3	97	Lacto-fermented		0.970	0.970	0.940
JRip (Rules)	96	4	Fresh	96.5	0.970	0.965	0.930
	3	97	Lacto-fermented		0.960	0.965	0.930

MCC—Matthews correlation coefficient.

**Table 3 foods-11-02956-t003:** The results of discrimination of fresh and lacto-fermented red bell pepper samples using selected image textures from color channel *L*.

Algorithm (Group)	Predicted Class (%)		Actual Class	Average Accuracy (%)	Precision	F-Measure	MCC
	Fresh	Lacto-Fermented					
IBk (Lazy)	98	2	Fresh	98	0.980	0.980	0.960
	2	98	Lacto-fermented		0.980	0.980	0.960
SMO (Functions)	98	2	Fresh	98	0.980	0.980	0.960
	2	98	Lacto-fermented		0.980	0.980	0.960
Random forest (Trees)	98	2	Fresh	98.5	0.990	0.985	0.970
	1	99	Lacto-fermented		0.980	0.985	0.970
Naïve Bayes (Bayes)	97	3	Fresh	97	0.970	0.970	0.940
	3	97	Lacto-fermented		0.970	0.970	0.940
Filtered classifier (Meta)	97	3	Fresh	97	0.970	0.970	0.940
	3	97	Lacto-fermented		0.970	0.970	0.940
JRip (Rules)	97	3	Fresh	96.5	0.960	0.965	0.930
	4	96	Lacto-fermented		0.970	0.965	0.930

MCC—Matthews correlation coefficient.

**Table 4 foods-11-02956-t004:** The discrimination of fresh and lacto-fermented red bell pepper samples based on models developed using selected image textures from color space RGB.

Algorithm (Group)	Predicted Class (%)		Actual Class	Average Accuracy (%)	Precision	F-Measure	MCC
	Fresh	Lacto-Fermented					
IBk (Lazy)	98	2	Fresh	98.5	0.990	0.985	0.970
	1	99	Lacto-fermented		0.980	0.985	0.970
SMO (Functions)	100	0	Fresh	99	0.980	0.990	0.980
	2	98	Lacto-fermented		1.000	0.990	0.980
Random forest (Trees)	96	4	Fresh	96.5	0.970	0.965	0.930
	3	97	Lacto-fermented		0.960	0.965	0.930
Naïve Bayes (Bayes)	99	1	Fresh	98.5	0.980	0.985	0.970
	2	98	Lacto-fermented		0.990	0.985	0.970
Filtered classifier (Meta)	94	6	Fresh	93	0.922	0.931	0.860
	8	92	Lacto-fermented		0.939	0.929	0.860
JRip (Rules)	91	9	Fresh	93.5	0.958	0.933	0.871
	4	96	Lacto-fermented		0.914	0.937	0.871

MCC—Matthews correlation coefficient.

**Table 5 foods-11-02956-t005:** The performance metrics of distinguishing fresh and lacto-fermented red bell pepper samples using selected image textures from color channel *R*.

Algorithm (Group)	Predicted Class (%)		Actual Class	Average Accuracy (%)	Precision	F-Measure	MCC
	Fresh	Lacto-Fermented					
IBk (Lazy)	98	2	Fresh	97.5	0.970	0.975	0.950
	3	97	Lacto-fermented		0.980	0.975	0.950
SMO (Functions)	99	1	Fresh	98.5	0.980	0.985	0.970
	2	98	Lacto-fermented		0.990	0.985	0.970
Random forest (Trees)	96	4	Fresh	97	0.980	0.970	0.940
	2	98	Lacto-fermented		0.961	0.970	0.940
Naïve Bayes (Bayes)	95	5	Fresh	96	0.969	0.960	0.920
	3	97	Lacto-fermented		0.951	0.960	0.920
Filtered classifier (Meta)	94	6	Fresh	91.5	0.895	0.917	0.831
	11	89	Lacto-fermented		0.937	0.913	0.831
JRip (Rules)	89	11	Fresh	90.5	0.918	0.904	0.810
	8	92	Lacto-fermented		0.893	0.906	0.810

MCC—Matthews correlation coefficient.

**Table 6 foods-11-02956-t006:** The discrimination performance metrics for models developed on the basis of selected image textures from color space XYZ for distinguishing fresh and lacto-fermented red bell pepper samples.

Algorithm (Group)	Predicted Class (%)		Actual Class	Average Accuracy (%)	Precision	F-Measure	MCC
	Fresh	Lacto-Fermented					
IBk (Lazy)	99	1	Fresh	99	0.990	0.990	0.980
	1	99	Lacto-fermented		0.990	0.990	0.980
SMO (Functions)	100	0	Fresh	99	0.980	0.990	0.980
	2	98	Lacto-fermented		1.000	0.990	0.980
Random forest (Trees)	97	3	Fresh	98	0.990	0.980	0.960
	1	99	Lacto-fermented		0.971	0.980	0.960
Naïve Bayes (Bayes)	98	2	Fresh	98	0.980	0.980	0.960
	2	98	Lacto-fermented		0.980	0.980	0.960
Filtered classifier (Meta)	95	5	Fresh	97	0.990	0.969	0.941
	1	99	Lacto-fermented		0.952	0.971	0.941
JRip (Rules)	95	5	Fresh	96.5	0.979	0.964	0.930
	2	98	Lacto-fermented		0.951	0.966	0.930

MCC—Matthews correlation coefficient.

**Table 7 foods-11-02956-t007:** The results of distinguishing fresh and lacto-fermented red bell pepper samples using selected image textures from color channel *X*.

Algorithm (Group)	Predicted Class (%)		Actual Class	Average Accuracy (%)	Precision	F-Measure	MCC
	Fresh	Lacto-Fermented					
IBk (Lazy)	99	1	Fresh	99	0.990	0.990	0.980
	1	99	Lacto-fermented		0.990	0.990	0.980
SMO (Functions)	100	0	Fresh	99	0.980	0.990	0.980
	2	98	Lacto-fermented		1.000	0.990	0.980
Random forest (Trees)	96	4	Fresh	97	0.980	0.970	0.940
	2	98	Lacto-fermented		0.961	0.970	0.940
Naïve Bayes (Bayes)	99	1	Fresh	98.5	0.980	0.985	0.970
	2	98	Lacto-fermented		0.990	0.985	0.970
Filtered classifier (Meta)	93	7	Fresh	95.5	0.979	0.954	0.911
	2	98	Lacto-fermented		0.933	0.956	0.911
JRip (Rules)	94	6	Fresh	93.5	0.931	0.935	0.870
	7	93	Lacto-fermented		0.939	0.935	0.870

MCC—Matthews correlation coefficient.

## Data Availability

The data presented in this study are available on request from the corresponding author.

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
