# Peer review of "The Changes in Bell Pepper Flesh as a Result of Lacto-Fermentation Evaluated Using Image Features and Machine Learning"

_foods, 2022, doi:10.3390/foods11192956_

Round 1
Reviewer 1 Report
This paper presents the result of a study discriminating changes in bell pepper flesh after fermentation. The topic fits the Journal.
However, the paper has many inherent flaws that make it clearly unacceptable for publication. I list the main aspects that require a complete re-work of the experiments and a different approach to model building, validation and testing.
1. The whole study is about one small batch of 20 bell peppers. This is insufficient for any advanced modeling technique, and certainly to be able to select the "best" model-feature combination.
2. The experimental design has many flaws. In fact there are only 20 (rather) independent samples, i.e. the individual bell peppers. The 5 pieces per pepper are not independent, and the 5 pieces of one pepper before and after treatment neither are.
3. Good practices in calibration - validation and testing are completely ignored, with the test set, where the actual performance is measured, being completely independent. There now only is calibration and cross-validation. And as I read it, it is random so that two pieces of one pepper can be in the calibration and validation sets, respectively. This is wrong.
Besides, there are many other aspects that are lacking e.g.
1. What are the textures that are extracted?
2. No picture of samples.
3. No reference method.
Author Response
Reviewer 1
This paper presents the result of a study discriminating changes in bell pepper flesh after fermentation. The topic fits the Journal.
However, the paper has many inherent flaws that make it clearly unacceptable for publication. I list the main aspects that require a complete re-work of the experiments and a different approach to model building, validation and testing.
- The whole study is about one small batch of 20 bell peppers. This is insufficient for any advanced modeling technique, and certainly to be able to select the "best" model-feature combination.
Answer: We are grateful to the Reviewer for this detailed comment. This is an incentive to take a closer look at the importance of the number of cases in classifying samples. In our study, the modeling was performed using 200 bell pepper cases. In total, 100 cases of fresh pepper and 100 cases of lacto-fermented pepper were used in the development of the models. 100 cases in one class were sufficient to divide the set into a training set and a test set and to train the model to achieve high correctness of the classification. In previous studies, 100 cases were also sufficient to obtain satisfactory results. However, we intend to carry out more in-depth research on this subject, taking into account different storage conditions and times, and a much larger number of cases. Therefore, we would like to thank the Reviewer for this comment and for drawing attention to the number of cases. Increasing the number of cases would also allow the application of deep learning to the classification of pepper samples, which may result in an increase in correctness. For example, Shahinfar et al. reported that increasing the number of training images from 10 to 1000 may result in improved performance metrics of classification. However, the increase in accuracy can be slight in some cases.
Shahinfar, S.; Meek, P.D.; Falzon, G. “How many images do I need?” Understanding how sample size per class affects deep learning model performance metrics for balanced designs in autonomous wildlife monitoring. Ecol. Inform. 2020, 57, 101085.
- The experimental design has many flaws. In fact there are only 20 (rather) independent samples, i.e. the individual bell peppers. The 5 pieces per pepper are not independent, and the 5 pieces of one pepper before and after treatment neither are.
Answer: All samples used before and after treatment were dependent. One hundred of the same samples were considered before and after lacto-fermentation as written in the manuscript: "The extracted fresh pepper pieces were subjected to imaging. The same pieces were also intended for spontaneous lacto-fermentation and then imaging as lacto-fermented forms.". After the fresh samples were imaged, the same samples were lacto-fermented. This ensures that the same 100 pieces (5 pieces of 20 fruits) were used before and after the treatment. Such experiment allowed for the dependence of the samples.
- Good practices in calibration - validation and testing are completely ignored, with the test set, where the actual performance is measured, being completely independent. There now only is calibration and cross-validation. And as I read it, it is random so that two pieces of one pepper can be in the calibration and validation sets, respectively. This is wrong.
Answer: Thank you very much for your careful reading of the manuscript and this comment. This point is explained in more detail in the revised manuscript. The analysis was carried out using the commonly used method of dividing the data into a test and validation set.
It has been explained as follows: “The dataset including a total of 200 cases was randomly divided into 10 parts. Each of the ten parts was considered in turn and treated as the test set. Whereas the remaining nine parts were regarded as the training sets. The learning was performed 10 times using different training sets. The result was determined as the average of 10 estimates. Ten folds are generally sufficient to obtain the best estimate [31].”
- Ian, H.W.; Eibe, F. Data Mining: Practical machine learning tools and techniques, 2nd ed.; Elsevier: 2005.
Besides, there are many other aspects that are lacking e.g.
- What are the textures that are extracted?
Answer: It has been added to the revised manuscript as Table 1.
Table 1. The selected texture parameters used to develop the models to distinguish fresh and lacto-fermented red bell pepper samples.
|
Color space Lab |
Color channel L |
Color space RGB |
Color channel R |
Color space XYZ |
Color channel X |
|
LHMean LHPerc01 LHPerc10 LHMaxm10 LS5SH1SumEntrp LS5SH3Correlat LS5SV5AngScMom LS4RZGLevNonU aS5SZ1Correlat aS4RNRLNonUni bHDomn01 bS5SN1AngScMom bS4RHShrtREmp bS4RNLngREmph |
LHMean LHPerc01 LHPerc10 LHPerc99 LHMaxm10 LS5SH1SumEntrp LS5SH3Correlat LS5SV5AngScMom LS4RZGLevNonU LS4RNRLNonUni |
RHPerc10 RHMaxm01 RHMaxm10 RSGVariance RS5SH1Correlat RS5SH3Correlat RS5SH3SumVarnc RS5SH5Correlat RS5SV5Correlat RS4RZGLevNonU RATeta1 GHKurtosis GHPerc01 GHPerc50 GHMaxm10 GS5SH1SumEntrp GS5SH3SumVarnc GS5SN5AngScMom GS4RNRLNonUni BHDomn10 |
RHPerc10 RHMaxm01 RHMaxm10 RS5SH1Correlat RS5SZ1DifVarnc RS5SH3Correlat RS5SN3AngScMom RS5SH5Correlat RS5SV5Correlat RS4RHGLevNonU RATeta1 |
XHPerc10 XHMaxm01 XHMaxm10 XS5SH1Correlat XS5SV1SumAverg XS5SH3SumEntrp XS5SZ3SumEntrp XS5SH5Correlat XS4RVGLevNonU XASigma YHDomn01 YHMaxm10 YS5SZ1SumEntrp YS4RNRLNonUni ZSGNonZeros |
XHPerc10 XHMaxm01 XHMaxm10 XS5SH1Correlat XS5SV1SumAverg XS5SH3SumEntrp XS5SZ3SumEntrp XS5SH5Correlat XS4RVGLevNonU XS4RNGLevNonU XASigma |
- No picture of samples.
Answer: The sample of red bell pepper fruit used in the experiments has been added as Figure 1. The images of fresh and lacto-fermented samples have been added as Figure 2.
- No reference method.
Answer: The methods of sampling, image acquisition and processing, the method of analysis of the obtained data, as well as the method of model development were carried out in accordance with standard procedures. Individual methods were previously used by the authors, but their combination for the evaluation of the changes in bell pepper flesh as a result of lacto-fermentation is innovative. Image analysis is a method that allows to obtain accurate and repeatable measurements. The authors tested many algorithms from various groups to classify the samples in an objective way. There was no need to use any additional methods.

Reviewer 2 Report
The manuscript is written with clear understanding of the project addressed. However, there are major concerns that need to be addressed to enhance the quality of the manuscript. My specific comments are as follows:
Introduction:
L33: “Consuming pepper can provide health benefits due to the presence of phytochemicals, including phenolic compounds, capsaicinoids…” Add citation
L40: “Green bell pepper contains chlorophylls and distinctive carotenoids…” Add citation
L41: “Red pepper is characterized by the presence of capsorubin and capsanthin. In the case of yellow pepper, violaxanthin, β-carotene, lutein, zeaxanthin…” Add citation
Based on your objectives, please compare how your study is different from those that have already been published
Materials and Methods:
Explain the process of ROI segmentation
There is not much discussion on model development using machine learning. How about split ratio of prediction, cross-validation, and validation dataset? Explain
Explain the methods on extraction of texture parameters
Results and discussion:
L295: “In the harvesting stage, the activities involving machine learning may be related to crop size, quality, skin color…” Add citation
Instead of mentioning the results, the authors should justify/explain the findings
Conclusions:
Justify the main finding of your study. Which one of the machine learning algorithms produce the best result?
Add recommendation for future studies.
General comments:
Please check the reference styles and grammar of the manuscript.
Author Response
Reviewer 2
The manuscript is written with clear understanding of the project addressed. However, there are major concerns that need to be addressed to enhance the quality of the manuscript. My specific comments are as follows:
Introduction:
L33: “Consuming pepper can provide health benefits due to the presence of phytochemicals, including phenolic compounds, capsaicinoids…” Add citation
Answer: Thank you very much for your valuable comment. It has been added as [1]
L40: “Green bell pepper contains chlorophylls and distinctive carotenoids…” Add citation
Answer: It has been added as [2]
L41: “Red pepper is characterized by the presence of capsorubin and capsanthin. In the case of yellow pepper, violaxanthin, β-carotene, lutein, zeaxanthin…” Add citation
Answer: It has been added as [2]
Based on your objectives, please compare how your study is different from those that have already been published
Answer: The justification of the novelty of our study has been expanded as follows:
“Despite the published results, mainly for other species of fruit and vegetables, there is no literature data on the use of texture parameters extracted from various col-or channels of images of cut red bell pepper to build classification models for monitoring the effect of lacto-fermentation on changes in the structure of pepper flesh. Therefore, the objective of this study was to propose an innovative approach to distinguishing fresh and lacto-fermented red bell pepper samples involving selected texture parameters of images and various machine learning algorithms. The application of textures from different color channels L, a, b, R, G, B, X, Y, and Z and algorithms belonging to different groups to evaluate the changes in red bell pepper flesh occurring as a result of spontaneous lacto-fermentation is a great novelty of this study. The innovative nature of the present study involves the acquisition of data on more than 1600 texture parameters of bell pepper flesh in the fresh and lacto-fermented forms. Various methods of selecting textures with the highest discriminant power, such as Genetic Search and Best First with the correlation-based feature selection (CFS), and the Ranker in conjunction with OneR attribute evaluator, were used to choose the most effective one. A wide range of algorithms from the groups of Lazy, Functions, Trees, Bayes, Meta, and Rules were tested which is not found in the literature data for the classification of fresh and processed bell pepper. The use of various algorithms in the discrimination process shows the validity, applicability and robustness of the proposed method rather than making a comparison between machine learning algorithms.”
Materials and Methods:
Explain the process of ROI segmentation
Answer: It has been explained as follows: “Firstly, the fresh and lacto-fermented bell pepper images were converted to color channels. In the case of lacto-fermented samples, changes in the structure of the flesh compared to the fresh samples are visible (Figure 2). The regions of interest (ROIs) were overlaid. ROI was considered as a set of pixels separated from the background. The segmentation of the image into lighter pepper pieces and the black background was performed using the brightness threshold which was determined manually. Each ROI was one whole piece of pepper. Thus, 200 ROIs including 100 ROIs of fresh samples and 100 ROIs of lacto-fermented samples were determined.”
There is not much discussion on model development using machine learning. How about split ratio of prediction, cross-validation, and validation dataset? Explain
Answer: It has been explained as follows: “The dataset including a total of 200 cases was randomly divided into 10 parts. Each of the ten parts was considered in turn and treated as the test set. Whereas the remaining nine parts were regarded as the training sets. The learning was performed 10 times using different training sets. The result was determined as the average of 10 estimates. Ten folds are generally sufficient to obtain the best estimate.”
Explain the methods on extraction of texture parameters
Answer: It has been explained as follows: “For each ROI, 1629 image textures were computed including 181 textures for each color channel. The texture parameters were computed based on the co-occurrence matrix (132 textures including 11 features for 4 directions and 3 between-pixels distances), run-length matrix (20 textures including 5 textures for 4 directions), Haar wavelet transform (10 textures), histogram (9 textures), autoregressive model (5 textures), gradient map (5 textures). Among the color channels of images selected for the texture ex-traction, color channels R (red), G (green), and B (blue) belonged to the RGB color space, channels L (lightness from black to white), a (green for negative and red for positive values), and b (blue for negative and yellow for positive values) - to the Lab color space and channels X (a component of color information), Y (lightness), and Z (a component of color information) - to the XYZ color space [28]. The computed image textures were considered as a function of the spatial variation of the pixel brightness intensity and provided information about the structure of the samples. Thus, the quantitative analysis of textures provided important insights into sample quality. The texture parameters after selection were used to build the discriminative models for distinguishing fresh and lacto-fermented red bell pepper samples.”
- Ibraheem, N.A.; Hasan, M.M.; Khan, R.Z.; Mishra, P.K. Understanding color models: a review. ARPN Journal of science and technology 2012, 2, 265-275.
Results and discussion:
L295: “In the harvesting stage, the activities involving machine learning may be related to crop size, quality, skin color…” Add citation
Answer: It has been added as [35]
Instead of mentioning the results, the authors should justify/explain the findings
Answer: It has been explained as follows: “When Table 2, Table 3, Table 4, Table 5, Table 6 and Table 7, which contain the results of this study, are examined, the findings regarding the experimental results can be expressed as follows: With image analysis, artificial intelligence and computer vision algorithms, fresh and lacto-fermented red bell pepper samples were distinguished in an objective, non-destructive and inexpensive way. In this way, the quality of the crop was determined automatically, quickly and without bias [34]. Also, the texture features of the different color channels used (L, a, b, R, G, B, X, Y, and Z) strongly represented the changes in lacto-fermented red bell pepper samples. It has been confirmed that these texture features can also be strongly distinguished by various machine learning algorithms from different groups (Lazy, Functions, Trees, Bayes, Meta, and Rules). According to the experimental results, the most erroneous discrimination was 90.5% with the Jrip algorithm, while the most successful discrimination was 99% with the IBk algorithm. Therefore, these results showed that the proposed method is highly preferable over conventional methods. Also, the results determined for red bell pepper in the present study confirmed the previous literature data on the effectiveness of models built based on image parameters using machine learning algorithms for the evaluation of the quality of fermented food products such as cucumber, carrot, or beetroot [22-25]. The undertaken research extended the scope of application of image analysis and artificial intelligence to the evaluation of fruit and vegetables by assessing the quality of fermented products.”
Conclusions:
Justify the main finding of your study. Which one of the machine learning algorithms produce the best result?
Answer: It has been indicated as follows: “Preservation and processing of bell peppers by lacto-fermentation is a traditional method to extend their shelf life. In this context, the main justification for this study is to evaluate the effect of the lacto-fermentation method on red bell peppers.
…
The highest results, including the average discrimination accuracy reaching 99% were obtained for the IBk from the group of Lazy, SMO (Functions), Random Forest (Trees), Naive Bayes (Bayes), Filtered Classifier (Meta), and JRip (Rules) machine learning algorithms.”
Add recommendation for future studies.
Answer: It has been indicated as follows: “… Due to the evaluation of the quality of lacto-fermented bell pepper using textures from different color channels L, a, b, R, G, B, X, Y, and Z and various machine learning algorithms belonging to different groups, the studies were characterized by novelty. The obtained results were very promising. Therefore, further research on the evaluation of the effect of lacto-fermentation on the flesh of other cultivars of pepper and other species of fruit or vegetables may be performed.
The texture features extracted from the different color channels strongly indicated the effect of lacto fermentation. However, a single texture feature may not be sufficient to distinguish lacto-fermented red bell pepper samples. The features extracted in machine learning algorithms determine the performance of the system. The features should therefore strongly represent the distinction between species. Therefore, in future studies, more than one texture feature will be fused, and the effect of lacto fermentation will be revealed more strongly. In addition, the new trend for agricultural discrimination tasks is to develop deep learning-based methods. This way, more powerful features are automatically extracted instead of manual feature extraction. Therefore, deep learning-based techniques can provide easier and more accurate discrimination. But to see the superior performance of deep learning clearly, more samples are required than machine learning. Therefore, in our next studies, the effect of lacto fermentation will be analyzed with deep learning-based CNN models that enable the extraction of high-level features. For this, datasets containing more samples will be prepared. Finally, the proposed study is only for bell pepper and therefore does not produce good results in differentiating different vegetables or fruits as a result of lacto fermentation. In this sense, it is planned to create a large dataset containing different fruits and vegetables in future studies.”
General comments:
Please check the reference styles and grammar of the manuscript.
Answer: The reference style and grammar have been corrected.

Reviewer 3 Report
1 Introduction
In general, the introduction raises the essential aspects of quality in paprika. However, several elements should be improved and presented in a new paper version.
The paragraphs are too long. It is suggested to separate it into small sections.
The authors present a problem. However, the solution is not a novelty since they are methods and algorithms used in other contexts. The scientific contribution is not defined. For example, using several analysis algorithms is not a scientific contribution.
The methodology used and the main results are missing.
There is a lack of a paragraph informing about the sections of the paper.
2 Materials and methods
2.1 materials
This section is very well explained.
Add examples of images of stored peppers.... at the beginning of the process and the end.
3 Results
Good regarding the results of the experiment.
Should add a discussion of the challenges of this type of work. Add a discussion regarding what other authors say for similar cases. It can be for the analysis of different vegetables or fruits.
The authors should explain the essential elements according to their expertise and compare them with the results obtained. Indicate possible trends with other technologies. Deep Learning is at the forefront. However, there are challenges, such as processing massive data and analyzing and storing information when there is volume.
In general, there is a need to present new results different from other cases. For example, analysis of images when peppers are together in a box or in complex situations to measure. The data processing should be able to identify each bell pepper to discriminate if any present conditions different from the normal ones.
Finally, the authors are requested to add a Background section that includes
Lacto-Fermentation Evaluated
Machine Learning algorithms
analysis process using ML
image processing
MaZda , explain with examples
Author Response
Reviewer 3
1 Introduction
In general, the introduction raises the essential aspects of quality in paprika. However, several elements should be improved and presented in a new paper version.
The paragraphs are too long. It is suggested to separate it into small sections.
Answer: In the introduction section, long paragraphs were separated into small sections. Please see the revised article.
The authors present a problem. However, the solution is not a novelty since they are methods and algorithms used in other contexts. The scientific contribution is not defined. For example, using several analysis algorithms is not a scientific contribution.
Answer: The scientific contributions and novelties of the proposed study are more clearly highlighted as follows.
“Despite the published results, mainly for other species of fruit and vegetables, there is no literature data on the use of texture parameters extracted from various col-or channels of images of cut red bell pepper to build classification models for monitoring the effect of lacto-fermentation on changes in the structure of pepper flesh. Therefore, the objective of this study was to propose an innovative approach to distinguishing fresh and lacto-fermented red bell pepper samples involving selected texture parameters of images and various machine learning algorithms. The application of textures from different color channels L, a, b, R, G, B, X, Y, and Z and algorithms belonging to different groups to evaluate the changes in red bell pepper flesh occurring as a result of spontaneous lacto-fermentation is a great novelty of this study. The innovative nature of the present study involves the acquisition of data on more than 1600 texture parameters of bell pepper flesh in the fresh and lacto-fermented forms. Various methods of selecting textures with the highest discriminant power, such as Genetic Search and Best First with the correlation-based feature selection (CFS), and the Ranker in conjunction with OneR attribute evaluator, were used to choose the most effective one. A wide range of algorithms from the groups of Lazy, Functions, Trees, Bayes, Meta, and Rules were tested which is not found in the literature data for the classification of fresh and processed bell pepper. The use of various algorithms in the discrimination process shows the validity, applicability and robustness of the proposed method rather than making a comparison between machine learning algorithms.”
The methodology used and the main results are missing.
Answer: The following paragraph has been added for the methodology used and the main result.
“This study offers an innovative and comprehensive approach to assessing the quality of bell pepper fruit. In this context, the supplied red bell pepper sample pieces are imaged before and after lacto-fermentation. Each of these images is converted to L, a, b, R, G, B, X, Y, and Z color channels. Using the brightness threshold, each sample is segmented from the background and regions of interest (ROI) are generated. Then, various texture features are extracted from ROI images with different color channels. Texture features extracted from different color channels show the effect of different textural information on the discrimination process. Feature selection algorithms are used for a large number of texture features obtained at this stage. Finally, these select-ed features are analyzed by various machine learning algorithms, resulting in highly accurate discrimination of fresh and lacto-fermented red bell pepper samples. The results show that the proposed method is capable of distinguishing with an accuracy of up to 99%.”
There is a lack of a paragraph informing about the sections of the paper.
Answer: A paragraph containing the chapter information of the paper was added to the revised article as follows.
“This paper is organized as follows. After the introduction, chapter 2 deals with the acquisition, imaging, processing, and statistical analysis of bell pepper samples. Chapter 3 includes the discrimination performances obtained as a result of evaluating the texture features extracted from different color channels with different machine learning algorithms. Finally, chapter 4 evaluates the proposed work and provides suggestions for future work”
2 Materials and methods
2.1 materials
This section is very well explained.
Add examples of images of stored peppers.... at the beginning of the process and the end.
Answer: The sample of red bell pepper fruit used in the experiments has been added as Figure 1. The images of fresh and lacto-fermented samples have been added as Figure 2.
3 Results
Good regarding the results of the experiment.
Should add a discussion of the challenges of this type of work. Add a discussion regarding what other authors say for similar cases. It can be for the analysis of different vegetables or fruits.
The authors should explain the essential elements according to their expertise and compare them with the results obtained. Indicate possible trends with other technologies. Deep Learning is at the forefront. However, there are challenges, such as processing massive data and analyzing and storing information when there is volume.
Answer: The limitations of the study and future study planning depending on these limitations are added to the revised article as follows.
“The texture features extracted from the different color channels strongly indicated the effect of lacto fermentation. However, a single texture feature may not be sufficient to distinguish lacto-fermented red bell pepper samples. The features extracted in machine learning algorithms determine the performance of the system. The features should therefore strongly represent the distinction between species. Therefore, in future studies, more than one texture feature will be fused and the effect of lacto fermentation will be revealed more strongly. In addition, the new trend for agricultural discrimination tasks is to develop deep learning-based methods. This way, more powerful features are automatically extracted instead of manual feature extraction. Therefore, deep learning-based techniques can provide easier and more accurate discrimination. But to see the superior performance of deep learning clearly, more samples are required than machine learning. Therefore, in our next studies, the effect of lacto fermentation will be analyzed with deep learning-based CNN models that enable the extraction of high-level features. For this, datasets containing more samples will be prepared. Finally, the proposed study is only for bell pepper and therefore does not produce good results in differentiating different vegetables or fruits as a result of lacto fermentation. In this sense, it is planned to create a large dataset containing different fruits and vegetables in future studies.”
In general, there is a need to present new results different from other cases. For example, analysis of images when peppers are together in a box or in complex situations to measure. The data processing should be able to identify each bell pepper to discriminate if any present conditions different from the normal ones.
Answer: The analysis was performed for all peppers in the same way. All images were acquired at the same parameters. No external conditions had a different impact on individual cases. It has been clearly specified in the revised manuscript as follows: “There were 20 pepper pieces in one image. In total, images of 100 pieces of fresh pepper and 100 pieces of lacto-fermented pepper were obtained. The image processing using the MaZda application.”, “In total, 100 cases of fresh pepper and 100 cases of lacto-fermented pepper were used in the development of the models. 100 cases in one class were sufficient to divide the set into a training set and a test set and to train the model to achieve high correctness of the classification.”, “The dataset including a total of 200 cases was randomly divided into 10 parts. Each of the ten parts was considered in turn and treated as the test set. Whereas the remaining nine parts were regarded as the training sets. The learning was performed 10 times using different training sets. The result was determined as the average of 10 estimates. Ten folds are generally sufficient to obtain the best estimate [31].”
Finally, the authors are requested to add a Background section that includes
Lacto-Fermentation Evaluated
Machine Learning algorithms
analysis process using ML
image processing
MaZda , explain with examples
Answer: This specified information has been added to the end of the Introduction section as follows.
“The effect caused by Lacto-fermentation on red bell peppers can be observed by an expert by conventional measurement and observation. However, with Agriculture 4.0, manual and cumbersome agricultural solutions are now being replaced by computer-based automatic systems. Nowadays, the application of non-invasive measuring techniques to evaluate external and internal fruit quality, e.g., involving image processing and artificial intelligence, is desired in the food industry [13]. Machine learning includes algorithms and tools needed to understand patterns within the data, extract important information from the data, and make decisions on a specific task by machines [14]. Thanks to computer vision systems, many precision agriculture applications such as monitoring of plant diseases, detection of pests, detection of fruit / vegetables, spraying, mapping, etc. are carried out automatically and quickly [15]. In order for these applications to be carried out automatically, the data obtained from different sensors is analyzed by a computer instead of an expert. However, most of the time, these data are not used directly and preprocessing is performed on the raw data. Image data is mostly obtained in agricultural applications, and in this sense, image processing techniques are of great importance. Preprocessing and feature extraction algorithms in image processing have been used in many agricultural applications [16]. With image preprocessing, the raw image is transformed into a suitable one for the desired task, and different steps such as thresholding, morphological operations, segmentation, etc. are applied to the raw image [17]. Then, different features related to texture, shape and color are extracted from the processed image with feature extraction steps [18]. These features are fed into learning algorithms to automatically provide the desired dis-crimination. In this context, artificial intelligence-based methods such as machine learning and deep learning are used to automatically distinguish different species based on these images or these features [19]. In agricultural discrimination applications, texture features are mostly used because of their strong discrimination ability [20]. MaZda, an open source C++ library, is a highly preferred software package as it allows both 2D and 3D image texture analysis and image preprocessing [21]. For this reason, it has been used in numerous agricultural applications. The change caused by lacto-fermentation can also be observed by analyzing textural features through MaZda. In addition, it is very important and necessary to automatically determine the effect of lacto-fermentation, as it is often applied for different fruits and vegetables. The image features and machine learning algorithms were successfully used in previous studies for evaluation of the changes, for example, in carrot [22], cucumber [23], and beetroot [24,25] caused by lacto-fermentation.”

Reviewer 4 Report
There are no pictures whatsoever in this paper. This is not acceptable and authors should add pictures of bell papers for make things more clear to the readers.
Author Response
There are no pictures whatsoever in this paper. This is not acceptable and authors should add pictures of bell papers for make things more clear to the readers.
Answer: Thank you very much for this comment. We totally agree with this opinion. The sample of red bell pepper fruit used in the experiments has been added as Figure 1. The images of fresh and lacto-fermented samples have been added as Figure 2.

Round 2
Reviewer 3 Report
Dear Authors
I have reviewed your work. You have indeed made all the requested changes.